



# Percent-level production of $^{40}$Ar by an overlooked mode of $^{40}$K decay

Jack Carter[1], Ryan B. Ickert[1,2], Darren F. Mark[1,3], Marrisa M. Tremblay[2], Alan J. Cresswell[1], David C.W. Sanderson[1]

[1]SUERC, Rankine Avenue, Scottish Enterprise Technology Park, East Kilbride, G75 0QF
[2]Department of Earth, Atmospheric, and Planetary Sciences, Purdue University, West Lafayette, IN 47901
[3] University of St Andrews, College Gate, St Andrews, KY16 9AJ, Fife, Scotland, UK

*Correspondence to*: J. Carter (j.carter.1@researach.gla.ac.uk)

**Abstract.** The decay of $^{40}$K to the stable isotopes $^{40}$Ca and $^{40}$Ar is used as a measure of time for both the K-Ca and K-Ar geochronometers, the latter of which is most generally utilized by the variant $^{40}$Ar/$^{39}$Ar system. The increasing precision of geochronology has forced practitioners to deal with the systematic uncertainties rooted in all radioisotope dating methods. A major component of these systematic uncertainties for the K-Ar and $^{40}$Ar/$^{39}$Ar techniques is imprecisely determined decay constants and an incomplete knowledge of the decay scheme of $^{40}$K. Recent studies question whether $^{40}$K can decay to $^{40}$Ar
via an electron capture directly to ground state (EC$_{ground}$), citing the lack of experimental verification as reasoning for its omission. In this study, we (1) provide a theoretical argument in favour of the presence of this decay mode, and (2) evaluate the magnitude of this decay mode by calculating the electron capture to positron ratio (EC$_{ground}$/$\beta^+$) and after combining it with other estimates, provide a best estimate of $175 \pm 65$ (2$\sigma$). We provide support for this calculation through comparison of the experimentally verified EC$_{ground}$/$\beta^+$ ratio of $^{22}$Na with our calculation using the theory of $\beta$ decay.  When combined with
measured values of $\beta^+$ and $\beta^-$ decay rates, this yields a partial decay constant for $^{40}$K direct to ground state $^{40}$Ar of $9.6 \pm 3.8 \times 10^{-13}$ a$^{-1}$ (2$\sigma$).  We calculate a partial decay constant of $^{40}$K to $^{40}$Ar of $0.590 \pm 0.014 \times 10^{-10}$ a$^{-1}$, total decay constant of $5.473 \pm 0.107 \times 10^{-10}$ a$^{-1}$ (2$\sigma$), and conclude that although omission of this decay mode can be significant for K-Ar dating, it is minor for $^{40}$Ar/$^{39}$Ar geochronology and is therefore unlikely to have significantly biased published measurements.

## 1. Introduction

$^{40}$K is a naturally occurring radioisotope of K with atomic abundance of 0.0117% (Garner et al., 1975). $^{40}$K undergoes a branched decay to $^{40}$Ar and $^{40}$Ca with a total half-life of ca. 1.3 Ga, and is the basis of the K-Ca and the K-Ar geochronometers (Aldrich and Nier, 1948; Wasserburg and Hayden, 1955; Marshall and DePaolo, 1982). The K-Ar system is most often exploited using the variant $^{40}$Ar/$^{39}$Ar method, wherein some of the $^{39}$K in the sample is transmuted to $^{39}$Ar by
irradiation with fast neutrons, thereby allowing both the parent and the daughter nuclides to be measured as isotopes of Ar (Merrihue and Turner, 1966). The latter is widely used to date geological events that span Earth history, from volcanic eruptions recorded in historical texts (e.g., Preece et al., 2018; Renne et al., 1997), to the earliest events in the solar system (e.g., Renne, 2000).



Advances in analytical precision have forced practitioners in geochronology to address systematic uncertainties that are inherent in all radioisotope dating methods, such as uncertainties in the measurement apparatus, prior assumptions made by the observer, or interference from environmental factors. For the K-Ar system, these uncertainties also include those that arise from imprecisely known decay rates of $^{40}$K. In the geological literature, there have been two influential reviews of measurements of the $^{40}$K decay rate. Beckinsale and Gale (1969) provided the first comprehensive review of measured and

predicted decay rates, which became the basis of the convention adopted by Steiger and Jäger (1977) used by the geochronological community for the next 20 years. Subsequently, Min et al. (2000) provided a more lengthy, critical review of available specific activity data determined by direct measurements of decay, and updated the derived decay rates for newer physical constants. More recently, the $^{40}$K decay parameters were estimated by Renne et al. (2010a,b), and although direct measurements of the $^{40}$K decay were incorporated into the estimate, it was heavily weighted to an intercomparison

with $^{238}$U decay. The decay rate determined by Renne et al. (2010, 2011), and the Min et al. (2000) decay rates are the most frequently used in $^{40}$Ar/$^{39}$Ar geochronology. These evaluations, along with those from the nuclear physics community, have been summarized recently by Cresswell et al. (2018, 2019).

Despite decades of work and longstanding interest in $^{40}$K decay, there remains uncertainty over the nature of the decay
scheme. There is consensus that most $^{40}$K decays by $\beta^-$ to $^{40}$Ca or by electron capture to $^{40}$Ar via an excited state, and that a small amount (~ 0.001%) of $^{40}$K decays to $^{40}$Ar via $\beta^+$. The early but influential review of $^{40}$K decay by Beckinsale and Gale (1969) included these decay modes, and also included a prediction of a second electron capture decay directly to the ground state of $^{40}$Ar that would add an additional ~2% to the rate of decay from $^{40}$K to $^{40}$Ar. Many subsequent workers both in nuclear physics and geochronology have ignored this prediction. The influential review by Min et al. (2000) described this
decay mode as "unverified" and having a "questionable" existence.

However, the putative electron capture to ground state decay mode decay constant is of the same order of magnitude as the uncertainties in the decay rate of $^{40}$K to $^{40}$Ar, and therefore may be a non-negligible and potentially important part of the geochronological system. Here, we describe the theoretical basis of this predicted decay mode and demonstrate the robust
nature of the prediction via an analogous calculation of $^{22}$Na decay. We describe experiments that could be made to measure this decay mode and also identify observations from nuclear physics experiments that offer evidence for its existence. We conclude that the evidence for this decay mode is strong, and despite the large uncertainty, should be considered in evaluations of the $^{40}$K decay rate.

## 2. Historical Overview

At present, $^{40}$K has three experimentally-verified decay modes (Figure 1):




1)  $\beta^-$ decay to $^{40}$Ca. This mode can be verified by direct measurement of the $\beta^-$ emission.

2)  Electron capture to an excited isomer of $^{40}$Ar, followed by decay to the ground state of $^{40}$Ar accompanied by emission of a 1.46 MeV γ-ray. Hereafter we denote this decay mode as EC*. This mode can be verified by direct measurement of the γ emission.

3)  $\beta^+$ decay from the ground state of $^{40}$K to the ground state of $^{40}$Ar (Engelkemeir et al., 1962). This is a very small component of the total decay rate and has been verified by direct measurement of the $\beta^+$ emission.

In their paper reporting the measurement of $\beta^+/\beta^-$, Engelkemier et al. (1962), through a private correspondence with Brosi and Kettle, proposed that an electron capture mode that goes directly to ground state $^{40}$Ar also exists, with an electron

capture to positron ratio of 155. This decay mode is hereafter denoted $EC_{ground}$.  This decay mode has not been experimentally detected, in part because the measurement is much more difficult to make than the others. If it exists, it would contribute about 0.2% to the total decay rate of $^{40}$K, or about 2% to the $^{40}$Ar branch.

The $EC_{ground}$ decay mode was included in the review by Beckinsale and Gale (1969) and then subsequently in Steiger and

Jäger (1977). This decay mode is also included in the widely-used ENSDF and DDEP evaluations (Chen, 2017 and Mougeot & Helmer, 2009 respectively). However, evaluations by Endt and Van der Leun (1973, 1978), Endt (1990), Audi et al., (2003) do not explicitly include this decay mode, with Audi et.al. (2003) giving a transition intensity which is the combined EC* and β⁺ intensities. Min et al., (2000) have questioned its validity because there is no experimental verification, and therefore do not include $EC_{ground}$ in their estimates.


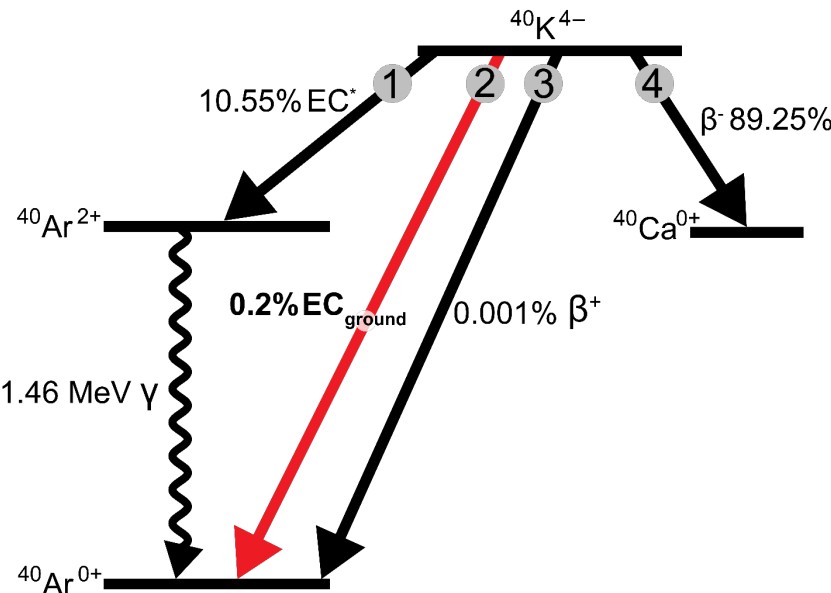

**Figure 1: Decay scheme of $^{40}$K after McDougall and Harrison (1999) and Leutz et al., (1965), where 1 is the electron capture branch to the excited state of $^{40}$Ar with γ-ray emission (EC\*), 2 is the electron capture direct to the ground state of $^{40}$Ar (EC$_{ground}$), 3 is the positron decay to ground state of $^{40}$Ar, and 4 is the β$^-$ decay to the ground state of $^{40}$Ca. The disputed decay mode, EC$_{ground}$,**
**is highlighted in red.**

### 3. Why there must be an EC$_{ground}$ decay mode

In nuclides that are too proton-rich and therefore radioactive, protons decay to correct this imbalance via two mechanisms. Either, (1) the nucleus undergoes electron capture wherein an orbital electron is captured by the nucleus, or (2) the nucleus emits a positron ($β^+$). Both processes are types of β decay and result in the transformation of a proton to a neutron to
conserve charge, and they both also emit a neutrino in order to conserve lepton number and energy. These two processes are typically paired: coupled electron capture-$β^+$ is the second most abundant decay type on the chart of the nuclides, after $β^-$ decay (Audi et al., 2003). They are linked because both processes have the same initial and final nuclear states.

$β^+$ decay is always accompanied by EC, but the converse is not always true (Bambynek et al., 1977). This is because $β^+$
decay, unlike EC, requires a minimum amount of energy (~1022 keV, equivalent to the combined rest masses of both a positron and an electron) in order to produce the emitted positron and an electron (the latter to satisfy charge conservation). In the decay of $^{40}$K, the EC\* branch has an energy difference between the initial and excited isomer state of only 44 keV. In contrast, the energy difference between $^{40}$K and the ground state of $^{40}$Ar, is 1504.4 keV (Wang et al., 2017), an energy





greater than the combined rest masses of the positron and electron. Therefore the EC* branch *cannot be* the complement to
the $\beta^+$ decay and the EC$_{ground}$ *must* exist to provide the $\beta^+$ complement.

### 4. Theory and Calculation of EC$_{ground}$/$\beta^+$

In the decay of $^{40}$K, the nuclide can reach a more stable state ($^{40}$Ca or $^{40}$Ar) only by violating quantum selection rules.
Decays which violate these selection rules undergo slow, so-called 'forbidden' unique transitions, which give $^{40}$K its long
~1.3 Ga half-life. The $^{40}$K decay scheme itself is unusual because the coupled EC$_{ground}$-$\beta^+$ and $\beta^-$ branches are the only third
order unique forbidden transitions known in nature.  All $^{40}$K decays undergo a parity reversal (where parity reversal is the
change of sign in one of the spatial coordinates *(x, y, z)*) between the initial parent state and final daughter state. Therefore
we can define the selection rules as:

$$'|\Delta J - 1|^{st} order\ unique\ forbidden\ decay\,'$$

where $\Delta J = J_i - J_f$, is the change in spin from initial to final state following Krane and Halliday (1987). We can
characterize each decay mode of $^{40}$K by its degree of forbiddenness from the above selection rule. The EC* mode undergoes
a spin change of $\Delta J = 4 - 2 = 2$ and is classified as a first order unique forbidden decay. The three other decay modes
of $^{40}$K, including EC$_{ground}$, all undergo a spin change of $\Delta J = 4 - 0 = 4$ and are classified as $3^{rd}$ order unique forbidden
decays.

The EC process occurs because the atomic electrons have a finite probability to be in the nucleus with the likelihood of being
captured highest for those closest to the nucleus. A theoretical description of     emission was first given by Fermi (1934),
while the possibility of electron capture which was first recognized by Yukawa and Sakata (1935) and later developed by
Bethe and Bacher (1936).  Here we use Fermi theory of $\beta^-$ decay to calculate the EC$_{ground}$/$\beta^+$ in the decay of $^{40}$K.

We can use the ratio of orbital electron capture and positron emission to infer the existence of EC$_{ground}$. The ratio *br* is
defined as:

$$br = \frac{\lambda_{ec}}{\lambda_{\beta^+}},\qquad\qquad\qquad\qquad\qquad (1)$$

Where $\lambda_{ec}$ and $\lambda_{\beta^+}$ are the probability per unit time of electron capture or $\beta^+$ emission. In electron capture, orbital electrons
can be captured from any orbital shell of the atom.  The EC/$\beta^+$ is therefore the summation of the individual capture ratios
from each shell.  Following Bambynek et al. (1977), the total electron capture-to-positron ratio is:

$$\frac{\lambda_x}{\lambda_{\beta^+}} = \frac{\sum_x n_x C_x f_x}{f_{\beta^+} C(W)},\qquad\qquad\qquad\qquad (2)$$



where $x$ is the shell, $n_x$ is the relative occupation number, $C_x$ contains the dependence of electron capture rates on nuclear structure giving the forbiddenness classification, similar to the shape factor in $\beta$ decay (Emery, 1975), $f_x$ is the integrated fermi function in $\beta$ decay, $f_{\beta+}$ is the integrated positron spectrum, and $C(W)$ is the theoretical shape factor for allowed or forbidden transitions. A review of shape factors for $^{40}$K transitions is provided by Cresswell et al. (2018, 2019). We initially simplify this equation to only consider the innermost K shell, the shell containing the electron with the highest probability to

be captured by the nucleus:

$$\frac{\lambda_k}{\lambda_{\beta+}} = \frac{n_k C_k f_k}{f_{\beta+} C(W)}, \tag{3}$$

where $\lambda_K$ is the probability of K-shell capture. For this capture, $f_K$ is defined as:

$$f_k = \frac{\pi}{2} q_k^2 \beta_k^2 B^k, \tag{4}$$

where $q_K$ is the momentum of the neutrino particle, $\beta_K$ is the Coulomb amplitude of the wave function, and $B_K$ is the term for

overlap and exchange corrections. Similarly, $f_{\beta+}$ is defined as:

$$f_{\beta+} = \int_1^{W_0} F(-Z,W) W p (W_0 - W)^2 dW, \tag{5}$$

$$W = 1 + \frac{E_T}{m_e}, \tag{6}$$

$$W_0 = 1 + \frac{E_{max}}{m_e}, \tag{7}$$

$$p = \sqrt{W^2 - 1}, \tag{8}$$

where W is the total energy of the positron given by its kinetic energy $E_T$ and rest mass $m_e$, defined above, and the momentum of the positron is given by p (eq. 8), $W_0$ is the total normalized energy defined above, and F(-Z,W) is the Fermi function. We follow Bambynek et al., (1977) in the formula for $\frac{C_K}{C(W)}$ which is given by:

$$\frac{C_K}{C(W)} = [(2L-1)!]^{-1} q_K^{2(L-1)} \left\{ \sum_{n=1}^{L} \lambda_n p^{2(n-1)} ((2n-1)! [2(L-n)+1]!)^{-1} \right\}^{-1}, \tag{9}$$

where L = ΔJ, and L = 1 for ΔJ = 0. The parameter $\lambda_n$ cannot be calculated in a straightforward manner and therefore we

follow a typical assumption that $\lambda_n = 1$ (Huber, 2011). This reduces the above expression to:

$$\frac{C_K}{C(W)} = \frac{q_K^6}{p^6 + q^6 + 7p^2 q^2 (p^2 + q^2)}, \tag{10}$$

In a given decay, the change in charge from the initial to final state can lead to an imperfect overlap of the wavefunctions of these states. Furthermore, given the indistinguishability of electrons, there is the possibility of an exchange effect wherein an electron does not necessarily come from the orbital where the vacancy appears. For instance, it is possible that a vacancy

may appear in the K-shell but the captured electron from an outer shell is then subsequently filled by the inner shell electron



(Bahcall, 1962; Bambynek et al., 1977). We follow Bahcall (1962) in implementing corrections for these effects, resulting in $B_K = 0.979$. Then using nuclear data given in Bambynek et al. (1977) we estimate an $EC_{ground}/\beta^+$ of 148.

We first note that this value is in approximate concordance with the private correspondence value in Engelkemier et al. (1962). However, this is only the capture ratio from the K-shell so we extend our model to a total electron capture ratio from

all orbitals following Bosch et al. (1977). The total electron-capture-to-positron ratio, an extension of Eq.1, is given by:

$$\frac{EC}{\beta^+} = \frac{K}{\beta^+}\left(1 + \frac{L}{K} + \frac{M}{L}\frac{L}{K} + \cdots\right), \tag{11}$$

We can simplify this equation by neglecting shells that make a negligible contribution. In $^{40}$K the probability of capture is dominated by the two inner shells K and L1, with approximate probability of ca. ~90% and ~10% with a negligible contribution from the shells further out. We can therefore omit all shell captures except K and L1 to arrive at the total

$EC_{ground}/\beta^+$ ratio:

$$\frac{EC}{\beta^+} = \frac{K}{\beta^+}\left(1 + \frac{L_1}{K}\right), \tag{12}$$

The ratio of each shell capture can be solved with the following equation:

$$\frac{x}{K} = \frac{\beta_x^2(W_0 - W_x)^2 B_x}{\beta_K^2(W_0 - W_x)^2 B_K}, \tag{13}$$

where x = L1 and the other symbols have the same definition as above. Using this equation we calculate a total $EC_{ground}/\beta^+$

of 164.

To further estimate the magnitude of the electron capture decay mode, we can perform another calculation of $EC_{ground}/\beta^+$ following Fireman (1949). This simplified form of the calculating $EC_{ground}/\beta^+$ is dependent only on the Q value (the difference between the initial and final state energies). This is given by:

$$\frac{\lambda_{ec}}{\lambda_{\beta^+}} = \frac{(\eta+2)^8}{0.450\eta^{6.5}}\frac{1}{0.0676 + 1.25\eta + 8.48\eta^2 + 12.5\eta^3 + 1.74\eta^4 + 0.079\eta^6}, \tag{14}$$

where $\eta = \frac{Q}{m_e} - 2$. We calculate an $EC_{ground}/\beta^+$ of 272 using this method and the updated Q-value of Wang et al. (2017). We note that despite discrepancies in these values for each method of evaluation, they are of the same order of magnitude. The differences in the values in these evaluations highlight the need for experimental measurement of $EC_{ground}/\beta^+$.






### 5. Comparison with other evaluations

Other theoretical evaluations of $EC_{ground}/\beta^+$ for $^{40}K$ exist in the literature (Figure 2). Pradler et al. (2013) and Mougeot (2018) report ratios of 150 and 212 ± 0.15, respectively (uncertainties are reported where they have been estimated). These workers

use broadly similar methods as us. Mougeot (2018) uses higher order corrections for both exchange and overlap and accounts for the dependence of $\lambda_{K,}$ that we set equal to 1 in Eq.9, on the energy of the decay. Pradler et al. (2013) use the Fermi method and data from Bambynek et al. (1977) but only perform the calculation for K-shell electrons, resulting in a slightly different calculated value than we report. Notably, all estimated values are of the same order of magnitude, similar to the ratio 155 reported in Engelkemier et al. (1962), and our calculated value of 164. Currently, the most commonly-used

$EC_{ground}/\beta^+$ value is calculated via the LogFT program, a program used in nuclear data evaluations (ENSDF Collaboration, LOGFT). However, the program is capable of only calculating first and second unique forbidden decay ratios, so the $EC_{ground}/\beta^+$ value from LogFT of 200 ± 100 is an extrapolation, with the assumption that the increase in the ratio from second to third order is by the same factor as the increase from first to second order. Finally, Chen (2017) evaluates the $^{40}K$ decay data and reports a $EC_{ground}/\beta^+$ value of 45.2 ± 1.4 without elaboration.


The variability between the modern estimates are driven primarily by more-or-less arbitrary choices when making the approximations necessary for these calculations to be tractable. Uncertainties on individual estimates which could be derived by propagating the uncertainties in the underlying experimental data are small – e.g., the estimated uncertainty provided by Mougeot (2018) is only ~0.1% (2σ) and is unlikely to capture the true uncertainty to which this quantity is

known.

Nevertheless, a recommended estimate and uncertainty is necessary for quantitative use. If we assume that the estimates are unbiased and approximately normally distributed, standard parametric statistics yield a mean and two standard deviation of the entire dataset of 192 ± 93. This value excludes the Chen (2017) evaluation ($EC_{ground}/\beta^+$ = 45.2 ± 1.4) as it is an extreme

outlier without further elaboration as to the methodology behind determining this value. If we exclude the oldest calculation (based on older fundamental data), the Chen (2017) value, and the two based on less sound theoretical underpinning (the log FT extrapolation and the estimate using the method of Fireman, 1949), the mean and two standard deviation are 175 ± 65. We propose the latter as the best current estimate of the $EC_{ground}/\beta^+$ ratio.



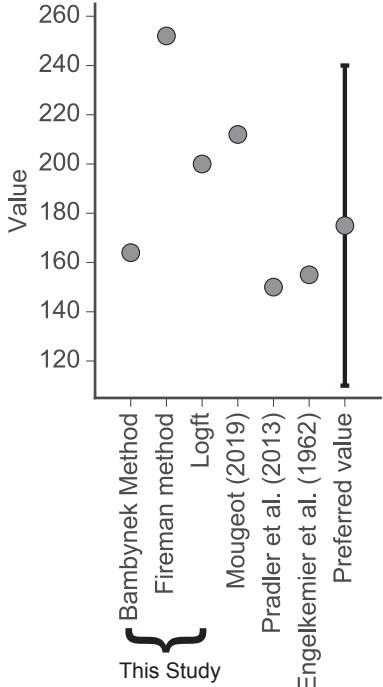


**Figure 2: Comparison of theoretically calculated $EC_{ground}/\beta^+$ of $^{40}K$ in this study using three methods; (1) Bambynek Method (Bambynek et al. (1977)), (2) Fireman method (Fireman, 1969), and (3) Logft (LogFT, 2001). The value of Chen (2017) is not included in the figure as it is an extreme outlier and the authors do not explain the method they use to reach this value. Our calculated ratios are compared to previous evaluations in the literature (Engelkemier et al. (1962); Pradler et al. (2013); Mougeot,**
**2019). Our preferred value, used in all calculations (175 ± 65) is also shown. Note the consistency in the estimated ratio from all of the methods is of the same order of magnitude, ~200.**

## 6. Comparison with $^{22}Na$

To test the validity of our $^{40}K$ $EC_{ground}/\beta^+$ estimate, we use the same calculations to estimate the experimentally-constrained $(EC/\beta^+)^*$ value for $^{22}Na$ decay. $^{22}Na$ is radionuclide with a half-life of ~2.6 years. It occurs in nature as a low-abundance
cosmogenic nuclide produced by spallation of $^{40}Ar$ and is also produced synthetically by proton irradiation for use in positron emission tomography. Like $^{40}K$, it decays by electron capture and positron emission. The main $EC$-$\beta^+$ pair for $^{22}Na$ decays initially to the excited state of $^{22}Ne$, followed by a 1.27 MeV $\gamma$ emission (Figure 3; Bé et al., 2006). This pair has a $(EC/\beta^+)^*$ of approximately 0.1 and accounts for >99.9% of the total decay. A second $EC$-$\beta^+$ pair decays directly to the ground state of $^{22}Na$ with an $(EC/\beta^+)_{ground}$ of ~ 0.02, but is a minor component. Here, we calculate the $(EC/\beta^+)^*$ for the main
branch. Unlike $^{40}K$, the dominant decay of $^{22}Na$ is the $\beta^+$ decay mode. This is due to the greater difference in energy between



the initial and final states, as positron decay will have a greater possibility of occurring in decays with a greater mass differences between initial and final states (Emery, 1975).

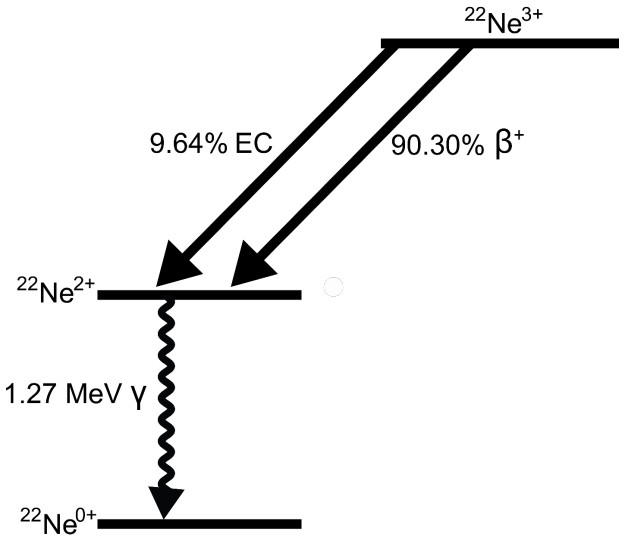

**Figure 3: Decay scheme of $^{22}$Na after Bé et al. (2006) and Leutz et al. (1965). An additional EC and β$^+$ decay pair that corresponds**
**to approximately 0.056% of the total decay of $^{22}$Na has been omitted for clarity.**

Unlike $^{40}$K, there are numerous measurements of the electron capture to positron ratio for decay to the excited state of $^{22}$Ne (Figure 4; Kreger, 1954; Vatai et al., 1968; Williams, 1964; McCann and Smith, 1969; Mac Mahon and Baerg, 1976; Bosch et al., 1977; Baerg, 1983; Schmidt-Ott et al., 1984; Sykora and Povinec, 1986; Kunze et al., 1990). Measurement of (EC/β$^+$)*
for $^{22}$Na is accomplished by measurement of both of the gammas (which come from both the EC* and the β$^{+*}$) and the x-rays (which only come from the EC branch). Relative to the $^{40}$K EC$_{ground}$/β$^+$, the $^{22}$Na (EC/β$^+$)* is easy to measure because of the higher activity (meaning higher count rate) and the higher energy of the x-ray emitted from the Auger electron, which an electron from the same atom that is emitted as a vacancy of an inner shell is filled. In a decay to the excited state of $^{22}$Ne, the de-excitation 1.28 MeV γ will be associated with both electron capture and positron decay. However, those measured 1.28
MeV γ that are not accompanied by two 0.511 MeV x-rays can be used to distinguish between both processes. We use the experimental measurements to verify our calculations described above for $^{40}$K.

Following a similar calculation using the Fermi method, our preferred method, to that used for our proposed estimate of the $^{40}$K EC$_{ground}$/β$^+$, we estimate an (EC/β$^+$)* of approximately 0.11. This is within the range of measured values of 0.105-0.115





(Fig. 4), suggesting that our calculation strategy of the the $^{40}$K $EC_{ground}/\beta^+$ is accurate, and lends further confidence to the existence of the current unmeasured $^{40}$K electron capture to ground state decay.

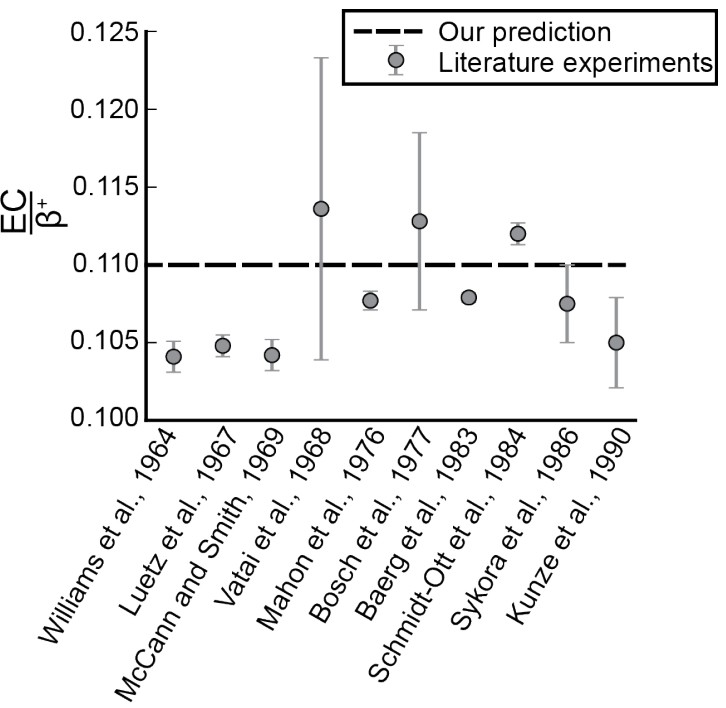

**Figure 4: Comparison of experimentally measured (EC/$\beta^+$)\* ratios of $^{22}$Na (grey circles) adapted from Kunze et al. (1990) with our**
**calculated value (black dashed line). Note the concordance of the theoretical and experimental determinations. The uncertainty in the Baerg et al. (1983) determination is smaller than the symbol.**

**7. Experimental verification of EC$_{ground}$ decay mode**

In both $\beta^-$ and $\beta^+$ decay, an electron or positron is emitted which allows for direct detection and verification of the decay process. In contrast, electron capture cannot be detected directly. Methods to experimentally verify electron capture rely on
indirect processes associated with the rearrangement of the atom following the capture of the orbital electron. Once the electron is captured the atom will rearrange itself to fill the vacancy, resulting in the emission of a characteristic x-ray or Auger electron with an energy defined by the binding energy of the shell vacancy of the daughter nucleus.

In the case of $^{40}$K, verification of the EC$_{ground}$ decay can be achieved by measuring the characteristic x-rays (Di Stefano et al., 2017). The orbital electron with the highest probability of capture is from the K-shell; if this electron is captured, it results in



the emission of a characteristic x-ray or Auger electron with an energy of 3.2029 keV, the binding energy of the K-shell of
$^{40}$Ar. However, electron capture to both the ground and excited state of $^{40}$Ar ($^{40}$Ar$^{2+}$) result in the same electron configuration
and x-ray emissions. Di Stefano et al. (2017) suggested tagging x-rays with the de-excitation γ associated with electron
capture to $^{40}$Ar$^{2+}$, which has a lifetime on the order of ~ 10$^{-12}$s (Di Stefano et al., 2017). X-rays tagged by the 1.46 MeV γ
must correspond to electron capture to the excited $^{40}$Ar$^{2+}$ state, with those x-rays not tagged correspond to the the electron

capture to ground state decay. Such an experiment will be challenging since it requires identifying a low probability decay
mode with x-ray signals present against a high background from the $^{40}$Ar$^{2+}$ state. The experiment therefore requires an x-ray
spectrometer able to resolve the Ar-K x-ray from other x-rays in the background, and accurately account for the x-ray – γ-ray
coincidence efficiency to quantify x-ray emission rates in excess of those from the $^{40}$Ar$^{2+}$ state. Given the complexity
involved in this experiment a pilot study was conducted at SUERC to meausre characteristic x-rays from a KCl source.  The

experiment was not successful because the detector was not able to resolve the Ar-K x-ray sufficiently, but demonstrates the
potential of this method to detect the x-rays, given a sufficiently high-resolution detector.  Full details are provided in the
supplementary material.

Ongoing attempts are being made to verify this decay mode by careful detection of the characteristic x-rays by the KDK

experiment (Di Stefano et al., 2017; Stukel, 2018). Experimental verification has implications for (1) rare event physics, as it
is a vital component in constraining the irreducible background and verifying results in the DArk MAtter (DAMA)
experiment (Pradler et al., 2013), (2) the theory of $\beta$- decay (Fermi, 1934) as it is the only 3$^{rd}$ order unique forbidden electron
capture decay known (Audi et al., 2003), and (3) K-Ar and $^{40}$Ar/$^{39}$Ar geochronology, for which it is currently overlooked due
to lack of experiment evidence. We further expand on the implications for geochronology below.

**8. Relevance for geochronology**

Geochronology with the K-Ar system requires either both the branching ratio and the total decay constant, or in the case of
an $^{40}$Ar/$^{39}$Ar age wherein the flux monitor age is constrained independently of its K-Ar systematics (Merrihue & Turner,
1966), only the total decay constant.  Using our preferred value of EC$_{ground}$/$\beta^+$ (175 ± 65, all uncertainties at 2σ), the decay
constants calculated by Min et al. ($\lambda_{EC*}$ = 0.580 ± 0.014 × 10$^{-10}$ a$^{-1}$ and $\lambda_T$= 5.463 ± 0.107 × 10$^{-10}$ a$^{-1}$), and the $\beta^+$/$\beta^-$  from

Engelkeimer et al. (1962) (1.12 ± 0.14 × 10$^{-5}$), we calculate a $\beta^+$ decay constant of 5.47 ± 0.69 × 10$^{-15}$ a$^{-1}$ and an EC$_{ground}$
decay constant of 9.6 ± 3.8 × 10$^{-13}$ a$^{-1}$.  Combining these values with the Min et al. (2000) values yields a new partial decay
constant for $^{40}$K to $^{40}$Ar of 0.590 ± 0.014 × 10$^{-10}$  a$^{-1}$, and total decay constant of 5.473 ± 0.107 × 10$^{-10}$ a$^{-1}$.  These values
include propagated uncertainties from our calculation and the Engelkeimer et al. (1962) $\beta^+$/$\beta^-$. However, the uncertainties
reported by Min et al. (2000) do not shift significantly due to the small size of the adjustment we propose. Existing and

modified constraints on the decay modes are given in Table 1.





**Table 1. Evaluations of decay mode branches and total decay constant used in age determination. $\lambda_{40Ar}$ is the partial decay constant for the $^{40}$Ar branch, including both the EC* and EC$_{ground}$ components.**

| Parameter | Value ± 2σ | Relative Unc. (%) | References |
|---|---|---|---|
| Previous values | | | |
| $\lambda_{EC*}$ | $0.580 \pm 0.014 \times 10^{-10}$ a$^{-1}$ | 2.4 | Min et al. (2000) |
| $\lambda_T$ | $5.463 \pm 0.107 \times 10^{-10}$ a$^{-1}$ | 2.0 | Min et al. (2000) |
| $\lambda_{\beta+}$ | $5.47 \pm 0.69 \times 10^{-15}$ a$^{-1}$ | 13 | Engelkeimer et al. (1962) |
| Modified values | | | |
| $\lambda_{ECground}$ | $9.6 \pm 3.8 \times 10^{-13}$ a$^{-1}$ | 40 | This work |
| $\lambda_{40Ar}$ | $0.590 \pm 0.014 \times 10^{-10}$ a$^{-1}$ | 2.4 | This work |
| $\lambda_T$ | $5.473 \pm 0.107 \times 10^{-10}$ a$^{-1}$ | 2.0 | This work |

Consequently, K-Ar (and $^{40}$Ar/$^{39}$Ar) ages calculated with these new decay constants will be younger than those calculated using the Min et al. (2000) decay constants. K-Ar dates are most sensitive to shifts in the decay constant because they incorporate the branching ratio, which is more strongly affected than the total $^{40}$K decay constant. K-Ar ages will decrease by 1.6% at 1 Ma, 1.3% at 1 Ga, and 0.7% at 4.5 Ga (Figure 5). Ages determined using the $^{40}$Ar/$^{39}$Ar method, for which the flux monitor age is independently constrained (e.g., Kuiper et al., 2008; Rivera et al., 2011), are much less sensitive to the

change in decay constant. Using equation 5 from Renne et al. (1998), and assuming calibration to a monitor with an age of 23.2 Ma, ages < 23 Ma increase only slightly, by < 0.002%. There is no age difference at 23.2 Ma, the flux monitor age. Ages then decrease for ages > 23.2 Ma, with ages decreased by 0.08% at 2.5 Ga, and by 0.11% at 4.5 Ga (Figure 5).



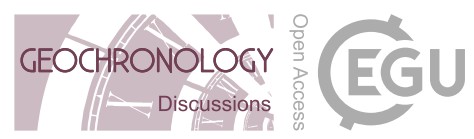

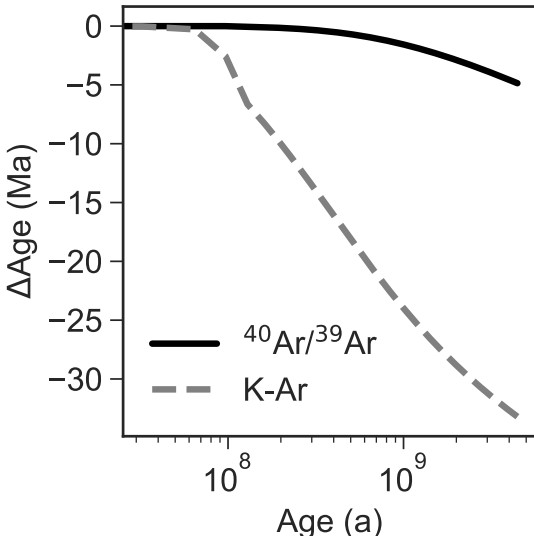

**Figure 5. Change in age, Δage,  is the age of a given sample with the decay mode to ground state included, subtracted from the age**

**with the decay mode to ground state omitted. The change in age using the K-Ar equation is shown in dashed grey (left axis) and**

**change in age using the $^{40}$Ar/$^{39}$Ar equation with independently calibrated standards is shown in solid black. The larger difference**

**in ages for the K-Ar system is due to the dependence on both the total decay constant and branching ratio.**

The age of flux monitors such as the Fish Canyon tuff sanidine (e.g., Morgan et al., 2014) determined by intercomparison

with astronomically tuned ages of ash beds (Kuiper et al., 2008; Rivera et al., 2011) is also sensitive to revision of decay

constants.  Using the data published by Kuiper et al. (2008), and incorporating an $EC_{ground}$ decay mode, we calculate a new

age for Fish Canyon sanidine of 28.200 ± 0.046 Ma, nominally lower, but indistinguishable from the published value of

28.201 ± 0.046 Ma. Overall, the effects of an $EC_{ground}$ decay mode are unlikely to be significant for most current applications

of $^{40}$Ar/$^{39}$Ar geochronology. However, given the levels of analytical precision attainable by the K-Ar dating approach when

dating geologically recent materials by K-Ar (e.g., Altherr et al. 2019), the $EC_{ground}$ decay mode will impact the accuracy of

this chronometer.



## 9. Conclusion

The Fermi theory of $\beta$ decay has decades of experimental support and is well established. We demonstrate this here by using these theories to accurately calculate the decay rate of a $^{22}$Na, a nuclide with an experimentally-verified decay rate. We have

used this information to demonstrate the high likelihood that the suspected second electron capture decay mode of $^{40}$K exists. We estimate the partial decay constant for $^{40}$K direct to ground state $^{40}$Ar to be $9.6 \pm 3.8 \times 10^{-13}$ $a^{-1}$ (2σ), based on combining multiple calculations with measurements of $\beta^-$ and $\beta^+$ decay rates. This addresses a longstanding question in K-Ar and $^{40}$Ar/$^{39}$Ar geochronology and provides future workers with confidence that the $^{40}$K EC$_{ground}$ decay mode exists. Just as important as providing support for its existence, we also demonstrate that the magnitude of this decay mode is small enough

that neglecting it has not yet resulted in significantly biased geochronological $^{40}$Ar/$^{39}$Ar data. The same cannot be stated for the K-Ar dating approach, especially for geologically-young materials.

Despite the strong grounding in theory, the EC$_{ground}$ decay mode has yet to be detected. The next step is experimental verification to determine the branching ratio. This will allow for a more complete evaluation of uncertainties associated with

the decay mode and the branching ratio. This experiment is difficult, but not intractable.

## 10. Author contribution

The study was conceived by JC and RBI. JC, AC, and DS calculated the ratio of electron capture to beta activities and measured x-rays. JC, RBI, DFM and MMT calculated the effects for geochronology. JC wrote the manuscript with contributions from all authors.

**Competing interests**

The authors declare that they have no conflicts of interest.

**Acknowledgements**

JC studentship funded by the UK Space Agency grant number ST/P001289/1. NERC are thanked for continued funding of the National Environmental Isotope Facility (Ar/Ar laboratory) at SUERC NE/S011587/1. MMT acknowledges The Royal

Society (NF171365) for funding. RBI thanks the members of the Geochron Club for discussion.





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
