# Peer review of "Production of 40Ar by an overlooked mode of 40K decay with implications for K-Ar geochronology"

_Geochronology, 2020_

## Referee Comment (RC1) · Anonymous Referee #1 · 29 Apr 2020

Comments on the manuscript entitled "Percent-level production for 40Ar by an overlooked mode of 40K decay" by Carter et al.

The paper discusses a potential electron-capture (EC) branch from the 40K ground state to the 40Ar ground state. Although the paper contains some interesting aspects, I cannot recommend accepting this article for publication. In the following I'd like to give a reasoning for my opinion.

1. The title is a bit misleading. The paper is dealing with a potential EC branch that contributes with 0.2% to all decays of 40K. Hence, it is not an additional "percent-level production of 40Ar". I also do not agree with "overlooked" since many works consider this potential EC branch (as I will also explain in the following).

[Figure]

2. The motivation of the article is – from my point of view – very weak. It is mainly based on the assertion that this branch is ignored or denied. This is not true. Considering nuclear decay data evaluations, nuclear physicists usually use only the most recent evaluation. Looking to two of the most important evaluation groups (ENSDF and DDEP), we find that an EC branch to the ground state was considered. Hence, the only remaining reference stated is Min et al. (2000). The authors do not mention several other publications which consider this branch. In line 53 they write "Many subsequent workers both in nuclear physics and geochronology have ignored this prediction." but do not provide references.

3. "Egelkemeir" (line 70) or Engelkemier (line 73 and Fig. 2, ..) or "Engelkeimer" (Table 1)? Just one example that indicates that the manuscript was not prepared with great care.

4. Several parts in the paper correspond to textbook knowledge in nuclear physics and could be omitted (e.g. large parts in section 4). Moreover, I am wondering whether the theoretical approach presented corresponds still to the state-of-the art. The theory from Bambynek (1977) was a standard for long time, but in the past $\sim$5 years, considerable progress was made by Prof. Mougeot in computation of beta (minus and plus) emission spectra and EC decay probabilities. I think his evaluation can be considered as state of the art. What is new or better in the paper under review? Note that the reference (Mougeot 2019) in Fig. 2 and in its caption should probably read "Mougeot (2018)".

5. The notations used are sometimes confusing and/or false. For example various expressions are used for the ratio of EC and beta+ decay probabilities. Line 162, the results refers only to K EC.

6. Is section 3 (in particular its title) justified? The reasoning is based on the assumption that a beta plus decay exists. If it exists, I agree that we may also expect an EC to ground state. However, the existence of the beta + with its very low probability in the order of 1E-5 is based on only few experiments. Did the authors consider that detected

positrons could also arise from internal pair production as in the decay of 90Y (i.e. from the beta minus side)? In this case, the whole reasoning would collapse.

7. Line 204: The authors question the uncertainty stated by Mougeot without providing any argument. At present, Mougeot is one person (perhaps the only) who can accurately calculate beta and EC decay with allowed and unique forbidden transitions.

8. Section 5: I cannot agree with the evaluation presented here. First, we must keep in mind that the results are correlated. Hence, a simple statistical consideration is not justified. Moreover, the choice of values that are taken into account appears to be very arbitrary. Most (actually all, except Mougeot) apply outdated models – as the authors do.

9. Section 6: Is it justified to use 22Na as cross check? The nature of the decay (allowed for the dominant transitions and 2nd forbidden unique for the others) are different than for 40K.

10. Section 6, Figure 4: There are more data. Why were they excluded? E.g. Applied Radiation and Isotopes, 66, 2008, 865-871 or Mougeot (2018) and Mougeot (2019) (Applied Radiation and Isotopes 154 (2019) 108884).

11. Line 241: "easy to meaure" I do not fully agree. Also 22Na is challenging, e.g. due to summing effects.

12. Line 245: x-ray with 511 keV? Perhaps "photon" or "annihilation photon". Also "gamma-ray" would be acceptable.

13. Line 264: "The orbital electron with the highest probability of capture is from the K-shell; if this electron is captured, it results in the emission of a characteristic x-ray or Auger electron with an energy of 3.2029 keV, the binding energy of the K-shell of 40Ar." No! After K-capture we have the K binding energy available. To eject an X-ray requires the binding energy of another electron in an outer shell. Hence, the x-ray energy is lower than the K binding energy. For Auger processed even 2 additional shells are

involved, which also means that x-rays and Auger electron do NOT have the same energy.

14. Line 269: "not tagged correspond to the the electron 270 capture to ground state decay". Really? But then you need 100% detection efficiency for the gamma-rays. "the the"?

15. Reference "Di Stefano et al.": List of authors is incomplete; Reference can be updated (Journal of Physics: Conference Series)

16. Line 60: "We describe experiments that could be made to measure this decay mode and also identify observations from nuclear physics experiments that offer evidence for its existence" I cannot find a sound description or proposal for such an experiment. Section 7 and the supplementary material are very weak. It is not clear how the two EC branches can be distinguished. What about the Auger contributions? If one could clearly identify x-rays as consequence of K EC, one would still need an x-ray emission probability. This is not even mentioned.

17. In general, the paper contains many formal errors and is not in compliance with ISO standard such as the GUM.

---

## Referee Comment (RC2) · Anonymous Referee #2 · 11 May 2020

I find this to be a valuable manuscript for geochronologists that provides evidence for the existence of a branch of the 40K decay system that has been questioned in the geochronology literature. Although the first referee indicates that the branch is not questioned in the nuclear physics community, this is not the case in the geochronology community, which in my opinion reasonably motivates the manuscript.

As a geochronologist, it is difficult for me to assess the validity of large sections of the paper (particularly section 4). Although the first referee states that parts of that section include 'textbook knowledge in nuclear physics', I find them valuable particularly given the submission of the manuscript to a geochronology-focused journal.

Specific Comments:

[Figure]

Line 70, Figure 1: In (for example) McDougall and Harrison (1999), the positron decay of 40K to 40Ar is shown as also involving a gamma ray at 1.02 MeV. However, Figure 1 shows the positron decay as direct to the ground state of 40Ar. Please address this discrepancy.

Line ~100: I'll preface this by noting that I'm not a physicist, but why doesn't the energy involved in the gamma ray come into account here? Why is it only the electron capture energy (and not the gamma) that matters? The total energy of the electron capture plus gamma would seem to be sufficient to couple with the positron decay. Also, given the note above (Line 70, Figure 1), is this argument affected by whether the positron decay goes directly to ground state or has an associated gamma?

Line 106: For readers who are not nuclear physicists, a brief explanation of quantum selection rules would make this more readable.

Figure 2 caption: Perhaps note that uncertainties were either not estimated or are smaller than the symbols? This is stated in the text but would be ideal to have in the caption as well.

Line 287 (and throughout): The use of 'flux monitor' is a common error – should be 'fluence monitor', as they are measuring the total neutron fluence (flux over time) affecting samples over the entire irradiation, rather than monitoring the neutron flux at one specific time.

Line 303: Based on Figure 5, I calculate a different percent decrease for K-Ar ages at 1 Ga (2.5%, for ca. 25 Ma at 1 Ga, rather 1.3%). The value 0.7% at 4.5 Ga seems accurate, and the 1.6% at 1 Ma is not identifiable (due to scaling of graph) in the figure. It would be helpful to have an expanded Figure 5, with multiple scales (or just expand this scale down to 1 Ma) to highlight different parts of the geologic timescale. It would also be helpful to show results in relative values as well as absolute values. Finally, the K-Ar line seems to have structure (e.g. around 10ˆ8 a) that should be explained.

Line 306: I'm struggling to understand the use of a fluence monitor at 23.2 Ma. I realize this is (may be?) a theoretical monitor, but it's in a paragraph with clear reference to Fish Canyon sanidine and I've spent some time wondering if '23.2' was a typo for '28.2 Ma'. Actually, I'm still not sure – is this a typo? If not, perhaps just note that it's a theoretical monitor of arbitrarily chosen age (if that's indeed what it is), to prevent others from wondering the same.

Technical Corrections:

Line 97: "They are linked because both processes have the same initial and final nuclear states." It's not clear what 'they' refers to – likely electron capture and positron, but perhaps beta?

Line 124: The symbol is missing from the pdf for type of emission

Line 148: Is Emax defined somewhere?

---

## Author Comment (AC1) · 29 Jul 2020

We would like to thank Anonymous Referee #1 (hereafter referred to as AR1) for their timely and detailed comments.

Overall, we disagree very little with the substance of AR1's review. However, we believe that there is a difference in perspective.

For better or for worse, the geochronological community compiles, calculates, and uses radioisotopic decay constants on its own, independent of the physics community. This is evidenced by the overwhelming use of the K decay constants published in the convention by Steiger and Jager (1977), Min et al. (2000), and to a lesser extent Renne et al. (2010) over any of the compilations favored by the

physics community. Indeed, the IUPAC and IUGS have commissioned a committee that publishes critical reviews of decay constants specifically for the chemical and geological community, that largely ignore the physics databases (Villa et al., 2015; Villa et al., 2016; Villa et al., in press; https://doi.org/10.1016/j.gca.2015.05.025; http://dx.doi.org/10.1016/j.gca.2015.10.011; https://doi.org/10.1016/j.gca.2020.06.022 ).

This manuscript is specifically targeted at geochronologists, who follow the conventions of geochronology, not of the physics literature. We agree with AR1 that our results are not sensu stricto new – a major point of the manuscript is that the original personal communication in 1962 is essentially correct. However, geochronologists do not typically follow the physics literature, are likely not to be well-trained in the type of nuclear physics that would recognize as obvious that there must be a complementary ground-state decay mode and are not qualified to judge the approximate magnitude. This manuscript makes this clear for geochronologists. That this is important should be clear from the way that decay constants are used by this community, and the literature from which they draw from. Essentially all 40Ar/39Ar and K-Ar geochronology draws on one of the three publications (Beckinsale and Gale 1969; Min et al., 2000; Renne et al., 2010/2011). Among these three, Min et al. and Renne et al. have the strongest underlying analysis and uncertainty structure (Min et al. effectively repeat the analysis embedded in Beckinsale and Gale, but more critically), and Min et al. is currently more popular than Renne et al. The confusion in the geochronological literature on the ground-state electron capture decay mode can be clearly traced to Min et al. That this is only a single paper does not weaken the argument for our manuscript, it strengthens it.

We strongly believe that there is a good place in the geochronological literature for our analysis; this is also supported by the comments in this regard from Anonymous Referee #2 (hereafter AR2). However, it is clear from the criticisms and comments of AR1 that some of our comments and data presentation may be misleading – we will
draft a revised manuscript that takes this feedback into account. These are responded to specifically below.

Response to specific points:

1) The "percent-level" change is percent of all decays to 40Ar, but we thank AR1 for pointing out that it may be misinterpreted and are happy to change the title of the manuscript. We disagree that it is not "overlooked", inasmuch as it is overlooked by the geochronological community, but we can also clarify this in a revised manuscript.

2) We state in the manuscript that the ground state branch is included in the ENSDF and DDEP evaluations. The fact that it is included in decay constant evaluations outside of those used by the geochronological community is important because it strengthens the argument that in favor of the presence of a EC ground state decay mode and should be considered (and is why they are discussed in our manuscript). However, simply because these evaluations exist, does not mean that our manuscript is not useful to the geochronolological community: the ENSDF and DDEP evaluations are not used for Ar-Ar or K-Ar dating.

The reference to "many subsequent workers" is somewhat confusingly written, and we thank AR1 for highlighting this. We are referring mainly to the many geochronological works that ignore this decay mode, but we do highlight some physics literature on line 82-83, and are also referring to the controversy over the DAMA experiment backgrounds, discussed on line 280. We will reframe this statement to be more clear and refer to specific literature.

3) We thank the reviewer for bringing this to our attention and the spelling will be corrected.

4) We are pleased that AR1 reads this section as an explanation of textbook knowledge. The audience for our manuscript is geochronologists, who are typically not versed in nuclear physics, so we have tried to provide a straightforward explanation

of the concepts that underpin this decay mode. The feedback from AR2 suggests that this was in fact useful for the geochronologist audience. We did not intend to represent our calculations as state of the art and will clarify this in the revised text by indicating that Mougeot (2018) provides the most robust calculation. Our purpose in providing additional calculations is to demonstrate to an audience of skeptical geochronologists (who may be skeptical of a decay constant that is derived entirely by calculation) that the derived quantity is relatively robust to differences in the way that it is estimated. For this reason we also include the much older Fireman (1949) calculation, and the extremely crude LogFT extrapolation, neither of which would be considered state of the art.

5) We thank AR1 for pointing out these inconsistencies and will fix them in a revised draft.

6) We thank AR1 for this observation. We have not considered that the single beta+ experiment might be erroneous, and we have taken the experiments at face value. If this experimental result is incorrect, all the physics literature that we and AR1 cite that includes the ground state decay mode is wrong. We rely on the Engelkemier et al. 1962 experimental observation of the positron decay, in this work the authors include a discussion of the pair production from 1462keV gamma in their experiment, calculating this as 55-60% of the total positron detection rate. The positrons produced from pair production would be mono-energetic at 440keV whereas the positron energy spectrum has an EMAX of 491 keV. We will include amended discussion points clarifying these points more clearly. It is outside of the scope of this manuscript to repeat the experiement of Engelkimeir et al. 1962, but we will gladly add the caveat in the revised manuscript that this hinges on a single measurement of a low probability decay mode.

7) We agree with AR1 that this is clumsily worded. The point we were trying to make is that the uncertainty budget for the estimate in Mougeot (2018) is not clearly articulated – it is not clear if the uncertainty presented in that paper is solely propagated from the Q-value as an intermediate precision, or whether it takes into account other sources of

uncertainty.

8) We take this criticism seriously from AR1. Ultimately, our goal with this manuscript is to provide a simple physics background and argument that will be straightforward to digest by working geochronologists, so that they understand the likely magnitude of a ground state E.C. decay of 40K. In our opinion, it is unlikely that a single estimate – even as cutting edge as possible – is likely to provide a convincing argument to our audience. For this reason, we have provided a range of estimates, using different techniques – yes, some outdated – that all point to a ECground/$\beta$+ of $\sim$200.

Most of the manuscript is dedicated to harnessing a range of evidence that it is likely that the decay mode exists and has a ECground/$\beta$+ of approximately 200. Having done this, our next goal was to then communicate the effect this has on the decay rate and branching ratio used in 40Ar/39Ar geochronology. For this, it's necessary to use an estimate, or several estimates.

We wanted to provide a reader, who for this journal is likely to be a geochronologist with skepticism that any one estimate will by unbiased or correct with a sense of what the possible range of effects is. The most straightforward way to do this is to group several reasonable estimates together and propagate that variance onto the existing decay rates used by the geochronological community (the values from Min et al.). The result is that the additional decay is negligible for most geochronological inference because it ends up being smaller than other uncertainties. Given that any reasonable ECground/$\beta$+ produces a negligible effect, we are not concerned that "averaging a number of different calculations" yields a result that was metrologically unsound, as AR1 argues.

It appears that the way we attempted to convey the overall magnitude of the effect of the non-zero ECground/$\beta$+ gives the wrong impression. We intend on revising the manuscript so that the focus is on the effect of range in calculated ECground/$\beta$+, and we will highlight and emphasize the Mougeot (2018) calculation as the preferred esti-

mate.

9) We agree that 22Na is not a perfect analog, but it is probably the best choice that has a tractable calculation and enough experimental data that the calculation can be reliably verified. In a revised manuscript we will clarify that this is not strictly analogous.

10) We regret overlooking Nähle et al. (2008), and will include this determination in a revised manuscript, thanks to AR1 for bringing this to our attention. We had not intended to present other calculations for Na22, but on the suggestion of AR1, we would be happy to include them.

11) In the manuscript, we meant it to be understood as "easier to measure than the 40K ground state decay", rather than imply that the 22Na decay is a straightforward measurement. We thank AR1 for pointing this out, and we will rephrase this section in a revised manuscript.

12) We will correct this statement in the revised manuscript.

13) We thank AR1 for identifying this confusing statement, and will correct it in a revised manuscript.

14) We thank AR1 for identifying this statement and we agree that by this wording is unclear. We will clarify this in a revised manuscript that will state that with a long enough counting period it will be possible to discern those x-rays that are untagged by the 1.46 MeV gamma and those that are tagged effectively observing the electron-capture to ground state decay mode.

15) We will correct this reference in a revised manuscript

16) ISO compliance is typically associated with certification and recertification of processes or products, and it is not clear how the ISO, or a particular ISO standard, is relevant to this manuscript. Regarding the Guide to Uncertainty in Measurement (GUM; JCGM 100:2008), the estimation of uncertainties follows what that guide refers to as the "law of propagation of uncertainty" described, for example, in section E.3.1

from the GUM. Our manuscript follows some geochemical and geochronological conventions that depart from GUM recommendations, such as notation and reference to "coverage factors" because this is the style of the journal and the community. These "formal errors" are largely an editorial decision, and we are happy to take direction from the editorial staff of Geochronology on this matter.

―――――――――――――――――

---

## Author Comment (AC2) · 29 Jul 2020

We would like to thank Anonymous Referee #2 (hereafter referred to as AR2) for their timely and detailed comments. We are encouraged that they identify as a geochronologist and find this to be a useful contribution to the literature. Overall, we are pleased that AR1 finds no overall fault with our conclusion that on balance, there is good physics-based evidence to support an ECground/$\beta \sim 200$, and that although AR2 identifies that they do not have the expertise to judge the physics argument, they believe it is a useful contribution to the geochronological literature.

Response to specific comments (numbered as they occur in RC2):

1) (line 70) The 1.02 MeV gamma is the sum of the 511 keV annhiliation photons from

the interaction of the positron with an electron. This is included in McDougall and Harrison (1999) adapted from Beckinsale and Gale (1969) in which it is not included. The 1.02 MeV will be an observation in any counting experiment however it is not a decay emission and as such we do not include it in Figure 1.

2) The energy that dictates if positron emission is possible is the Q value. The Q value is the difference between the initial mass state and final mass product. This energy is shared between the outgoing neutrino, atomic excitation of the daughter system, recoil energy, and possibly nuclear excitation of the daughter system. The Q-value therefore includes the excitation of the daughter system. However, positron decay can only compete with the electron capture if the Q value of the electron capture decay itself is greater than the threshold 1022 MeV value requiring the positron decay to go directly to the ground state. We will amend this to provide a clear statement in a revised manuscript.

3) Quantum selection rules place formal constraints of the possible transitions of a system from one quantum state to another. In our case it places constraints on the possible set of transitions from the parent 40K state to the daughter 40Ar. We will include this definition in the revised manuscript.

4) (Figure 2 caption) We will indicate in the caption that the uncertainties are either unknown or too small to plot.

5) (Line 287) We will correct flux monitor to fluence monitor.

6) (line 303) For a 40Ar/40K = 0.08, and using the decay constants in our Table 1 ($\lambda EC^*$ = 0.580e-10 and $\lambda$total = 5.463e-10 for Min et al; and $\lambda EC^*$ = 0.590e-10 and $\lambda$total = 5.473e-10; ), we used the K-Ar equation:

time = 1/$\lambda$total*ln(1+ $\lambda$total/$\lambda$Ar or EC**40Ar/40K).

This yields dates of 1028.05 Ma and 1014.24 Ma for the Min et al. and our revised decay constants, respectively. This is a difference of 13.81 Ma, or about 1.3 %. We
have reproduced this calculation and believe that our original calculation is correct, though we welcome any correction from AR2 if we have misunderstood.

In a revised manuscript, we will provide a figure that provides more detail. The small kink at 100 Ma is an Illustrator artifact and will be smoothed in a revised manuscript.

7) (line 306) We will change the date to 28.2 Ma.

Technical Corrections:

8) (Line 97) "They" refers to EC and positron, and will be clarified in the revised text.

9) (line 124) The missing quantity is a beta, and we will ensure this is typeset correctly in a revised text.

10) (line 148) Unfortunately we neglected to define E_max, but will do so in a revised text.

---

## Author Response (AR2)

**Response to reviewers**

Reviewer 1

We would like to thank Anonymous Referee #1 (hereafter referred to as AR1) for their timely and detailed comments.

Overall, we disagree very little with the substance of AR1's review. However, we believe that there is a difference in perspective.

For better or for worse, the geochronological community compiles, calculates, and uses radioisotopic decay constants on its own, independent of the physics community. This is evidenced by the overwhelming use of the K decay constants published in the convention by Steiger and Jager (1977), Min et al. (2000), and to a lesser extent Renne et al. (2010) over any of the compilations favored by the physics community. Indeed, the IUPAC and IUGS have commissioned a committee that publishes critical reviews of decay constants specifically for the chemical and geological community, that largely ignore the physics databases (Villa et al., 2015; Villa et al., 2016; Villa et al., in press; https://doi.org/10.1016/j.gca.2015.05.025; http://dx.doi.org/10.1016/j.gca.2015.10.011; https://doi.org/10.1016/j.gca.2020.06.022 ).

This manuscript is specifically targeted at geochronologists, who follow the conventions of geochronology, not of the physics literature. We agree with AR1 that our results are not *sensu stricto* new – a major point of the manuscript is that the original personal communication in 1962 is essentially correct. However, geochronologists do not typically follow the physics literature, are likely not to be well-trained in the type of nuclear physics that would recognize as obvious that there must be a complementary ground-state decay mode and are not qualified to judge the approximate magnitude. This manuscript makes this clear *for geochronologists*.

That this is important should be clear from the way that decay constants are used by this community, and the literature from which they draw from. Essentially all $^{40}Ar/^{39}Ar$ and K-Ar geochronology draws on one of the three publications (Beckinsale and Gale 1969; Min et al., 2000; Renne et al., 2010/2011). Among these three, Min et al. and Renne et al. have the strongest underlying analysis and uncertainty structure (Min et al. effectively repeat the analysis embedded in Beckinsale and Gale, but more critically), and Min et al. is currently more popular than Renne et al.

The confusion in the geochronological literature on the ground-state electron capture decay mode can be clearly traced to Min et al. That this is only a single paper does not weaken the argument for our manuscript, it strengthens it.

We strongly believe that there is a good place in the geochronological literature for our analysis; this is also supported by the comments in this regard from Anonymous Referee #2 (hereafter AR2). However, it is clear from the criticisms and comments of AR1 that some of our comments and data presentation may be misleading – we will draft a revised manuscript that takes this feedback into account. These are responded to specifically below.

Response to specific points:

1) *The title is a bit misleading. The paper is dealing with a potential EC branch that contributes with 0.2% to all decays of 40K. Hence, it is not an additional "percent-level production of 40Ar". I also do not agree with "overlooked" since many works consider this potential EC branch (as I will also explain in the following).*

The "percent-level" change is percent of all decays to $^{40}$Ar, but we thank AR1 for pointing out that it may be misinterpreted and are happy to change the title of the manuscript. We disagree that it is not "overlooked", inasmuch as it is overlooked by the geochronological community, but we can also clarify this in a revised manuscript.

2) *The motivation of the article is – from my point of view – very weak. It is mainly based on the assertion that this branch is ignored or denied. This is not true. Con- sidering nuclear decay data evaluations, nuclear physicists usually use only the most recent evaluation. Looking to two of the most important evaluation groups (ENSDF and DDEP), we find that an EC branch to the ground state was considered. Hence, the only remaining reference stated is Min et al. (2000). The authors do not mention several other publications which consider this branch. In line 53 they write "Many subsequent workers both in nuclear physics and geochronology have ignored this prediction." but do not provide references.*

(line 69 -71) We state in the manuscript that the ground state branch is included in the ENSDF and DDEP evaluations. The fact that it is included in decay constant evaluations outside of those used by the geochronological community is important because it strengthens the argument that in favor of the presence of a EC ground state decay mode and should be considered (and is why they are discussed in our manuscript). However, simply because these evaluations exist, does not mean that our manuscript is not useful to the geochronolological community: the ENSDF and DDEP evaluations are not used for Ar-Ar or K-Ar dating. The reference to "many subsequent workers" is somewhat confusingly written, and we thank AR1 for highlighting this. We are referring mainly to the many geochronological works that ignore this decay mode, but we do highlight some physics literature on line 82-83, and are also referring to the controversy over the DAMA experiment backgrounds, discussed on line 280. We will reframe this statement to be more clear and refer to

specific literature.

*3)* *"Egelkemeir" (line 70) or Engelkemier (line 73 and Fig. 2, ..) or "Engelkeimer" (Table 1)? Just one example that indicates that the manuscript was not prepared with great care*

We thank the reviewer for bringing this to our attention and the spelling will be corrected.

*4)* *Several parts in the paper correspond to textbook knowledge in nuclear physics and could be omitted (e.g. large parts in section 4). Moreover, I am wondering whether the theoretical approach presented corresponds still to the state-of-the art. The theory from Bambynek (1977) was a standard for long time, but in the past ~5 years, considerable progress was made by Prof. Mougeot in computation of beta (minus and plus) emission spectra and EC decay probabilities. I think his evaluation can be considered as state of the art. What is new or better in the paper under review? Note that the reference (Mougeot 2019) in Fig. 2 and in its caption should probably read "Mougeot (2018)".*

We are pleased that AR1 reads this section as an explanation of textbook knowledge. The audience for our manuscript is geochronologists, who are typically not versed in nuclear physics, so we have tried to provide a straightforward explanation of the concepts that underpin this decay mode. The feedback from AR2 suggests that this was in fact useful for the geochronologist audience. We did not intend to represent our calculations as state of the art and will clarify this in the revised text by indicating that Mougeot (2018) provides the most robust calculation. Our purpose in providing additional calculations is to demonstrate to an audience of skeptical geochronologists (who may be skeptical of a decay constant that is derived entirely by calculation) that the derived quantity is relatively robust to differences in the way that it is estimated. For this reason we also include the much older Fireman (1949) calculation, and the extremely crude LogFT extrapolation, neither of which would be considered state of the art.

*5)* *The notations used are sometimes confusing and/or false. For example various expressions are used for the ratio of EC and beta+ decay probabilities. Line 162, the results refers only to K EC.*

We thank AR1 for pointing out these inconsistencies and will fix them in a revised draft.

6) *Is section 3 (in particular its title) justified? The reasoning is based on the assump- tion that a beta plus decay exists. If it exists, I agree that we may also expect an EC to ground state. However, the existence of the beta + with its very low probability in the order of 1E-5 is based on only few experiments. Did the authors consider that detected positrons could also arise from internal pair production as in the decay of 90Y (i.e. from the beta minus side)? In this case, the whole reasoning would collapse.*

We thank AR1 for this observation. We have not considered that the single beta+ experiment might be erroneous, and we have taken the experiments at face value. If this experimental result is incorrect, all the physics literature that we and AR1 cite that includes the ground state decay mode is wrong. It is outside of the scope of this manuscript to redo the experiment, but we will gladly add the caveat in the revised manuscript that this hinges on a single measurement of a low probability decay mode.

7) *Line 204: The authors question the uncertainty stated by Mougeot without provid- ing any argument. At present, Mougeot is one person (perhaps the only) who can accurately calculate beta and EC decay with allowed and unique forbidden transitions.*

(line 215 – 217) We agree with AR1 that this is clumsily worded. The point we were trying to make is that the uncertainty budget for the estimate in Mougeot (2018) is not clearly articulated – it is not clear if the uncertainty presented in that paper is solely propagated from the Q-value as an intermediate precision, or whether it takes into account other sources of uncertainty.

8) *Section 5: I cannot agree with the evaluation presented here. First, we must keep in mind that the results are correlated. Hence, a simple statistical consideration is not justified. Moreover, the choice of values that are taken into account appears to be very arbitrary. Most (actually all, except Mougeot) apply outdated models – as the authors do*

We take this criticism seriously from AR1. Ultimately, our goal with this manuscript is to provide a simple physics background and argument that will be straightforward to digest by working geochronologists, so that they

understand the likely magnitude of a ground state E.C. decay of $^{40}$K. In our opinion, it is unlikely that a single estimate – even as cutting edge as possible – is likely to provide a convincing argument to our audience. For this reason, we have provided a range of estimates, using different techniques – yes, some outdated – that all point to a $EC_{ground}/\beta^+$ of ~200.

Most of the manuscript is dedicated to harnessing a range of evidence that it is likely that the decay mode exists and has a $EC_{ground}/\beta^+$ of approximately 200. Having done this, our next goal was to then communicate the effect this has on the decay rate and branching ratio used in $^{40}$Ar/$^{39}$Ar geochronology. For this, it's necessary to use an estimate, or several estimates.

We wanted to provide a reader, who for this journal is likely to be a geochronologist with skepticism that any one estimate will by unbiased or correct with a sense of what the possible range of effects is. The most straightforward way to do this is to group several reasonable estimates together and propagate that variance onto the existing decay rates used by the geochronological community (the values from Min et al.). The result is that the additional decay is negligible for most geochronological inference because it ends up being smaller than other uncertainties. Given that any reasonable $EC_{ground}/\beta^+$ produces a negligible effect, we are not concerned that "averaging a number of different calculations" yields a result that was metrologically unsound, as AR1 argues.

It appears that the way we attempted to convey the overall magnitude of the effect of the non-zero $EC_{ground}/\beta^+$ gives the wrong impression. We intend on revising the manuscript so that the focus is on the effect of range in calculated $EC_{ground}/\beta^+$, and we will highlight and emphasize the Mougeot (2018) calculation as the preferred estimate.

9) *Section 6: Is it justified to use 22Na as cross check? The nature of the decay (al- lowed for the dominant transitions and 2nd forbidden unique for the others) are different than for 40K.*

We agree that 22Na is not a perfect analog, but it is probably the best choice that has a tractable calculation and enough experimental data that the calculation can be reliably verified. In a revised manuscript we will clarify that this is not strictly analogous.

10) *Section 6, Figure 4: There are more data. Why were they excluded? E.g. Applied Radiation and Isotopes, 66, 2008, 865-871 or Mougeot (2018) and Mougeot (2019) (Applied Radiation and Isotopes 154 (2019) 108884).*

(Figure 4) We regret overlooking Nähle et al. (2008), and will include this determination in a revised manuscript,

thanks to AR1 for bringing this to our attention. We had not intended to present other calculations for Na22, but on the suggestion of AR1, we would be happy to include them.

*11) Line 241: "easy to meaure" I do not fully agree. Also 22Na is challenging, e.g. due to summing effects.*

(line 248) In the manuscript, we meant it to be understood as "easier to measure than the $^{40}$K ground state decay", rather than imply that the $^{22}$Na decay is a straightforward measurement. We thank AR1 for pointing this out, and we will rephrase this section in a revised manuscript.

*12) Line 245: x-ray with 511 keV? Perhaps "photon" or "annihilation photon". Also "gamma-ray" would be acceptable.*

(line 252) We will correct this statement in the revised manuscript.

*13) Line 264: "The orbital electron with the highest probability of capture is from the K-shell; if this electron is captured, it results in the emission of a characteristic x-ray or Auger electron with an energy of 3.2029 keV, the binding energy of the K-shell of 40Ar." No! After K-capture we have the K binding energy available. To eject an X-ray requires the binding energy of another electron in an outer shell. Hence, the x-ray energy is lower than the K binding energy. For Auger processed even 2 additional shells are involved, which also means that x-rays and Auger electron do NOT have the same energy.*

(line 273 – 279)  We thank AR1 for identifying this confusing statement, and will correct it in a revised manuscript.

*14) Line 269: "not tagged correspond to the the electron 270 capture to ground state decay". Really? But then you need 100% detection efficiency for the gamma-rays. "the the"?*

 (lines 283 - 285) We thank AR1 for identifying this statement and we agree that by this wording is unclear. We will clarify this in a revised manuscript that will state that with a long enough counting period it will be possible to discern those x-rays that are untagged by the 1.46 MeV gamma and those that are tagged effectively observing the electron-capture to ground state decay mode.

*15) Reference "Di Stefano et al.": List of authors is incomplete; Reference can be updated (Journal of Physics: Conference Series)*

We will correct this reference in a revised manuscript.

*16) Line 60: "We describe experiments that could be made to measure this decay mode and also identify observations from nuclear physics experiments that offer evi- dence for its existence" I cannot find a sound description or proposal for such an ex- periment. Section 7 and the supplementary material are very weak. It is not clear how the two EC branches can be distinguished. What about the Auger contributions? If one could clearly identify x-rays as consequence of K EC, one would still need an x-ray emission probability. This is not even mentioned.*

We will correct this statement in a revised manuscript that state we are only carrying out a simple test with the equipment available at SUERC to attempt to detect the ~3keV x-rays associated with both electron capture decay braches of $^{40}$K.

*17) In general, the paper contains many formal errors and is not in compliance with ISO standard such as the GUM.*

ISO compliance is typically associated with certification and recertification of processes or products, and it is not clear how the ISO, or a particular ISO standard, is relevant to this manuscript. Regarding the Guide to Uncertainty in Measurement (GUM; JCGM 100:2008), the estimation of uncertainties follows what that guide refers to as the "law of propagation of uncertainty" described, for example, in section E.3.1 from the GUM. Our manuscript follows some geochemical and geochronological conventions that depart from GUM recommendations, such as notation and reference to "coverage factors" because this is the style of the journal and the community. These "formal errors" are largely an editorial decision, and we are happy to take direction from the editorial staff of *Geochronology* on this matter. We do agree that our uncertainties for this value are misleading in the revised manuscript we will use lower and upper bounds of this value and propagate these through the decay constant determined by Min et al. (2000).

We would like to thank Anonymous Referee #2 (hereafter referred to as AR2) for their timely and detailed comments. We are encouraged that they identify as a geochronologist and find this to be a useful contribution to the literature. Overall, we are pleased that AR1 finds no overall fault with our conclusion that on balance, there is good physics-based evidence to support an $EC_{ground}/\beta \sim 200$, and that although AR2 identifies that they do not have the expertise to judge the physics argument, they believe it is a useful contribution to the geochronological literature.

Response to specific comments (numbered as they occur in RC2)

1)  *Line 70, Figure 1: In (for example) McDougall and Harrison (1999), the positron decay of 40K to 40Ar is shown as also involving a gamma ray at 1.02 MeV. However, Figure 1 shows the positron decay as direct to the ground state of 40Ar. Please address this discrepancy.*

 (line 70) The 1.02 MeV gamma is the sum of the 511 keV annhiliation photons from the interaction of the positron with an electron. This is included in McDougall and Harrison (1999) adapted from Beckinsale and Gale (1969) in which it is not included. The 1.02 MeV will be an observation in any counting experiment however it is not a decay emission and as such we do not include it in Figure 1.

2)  *Line ~100: I'll preface this by noting that I'm not a physicist, but why doesn't the energy involved in the gamma ray come into account here? Why is it only the electron capture energy (and not the gamma) that matters? The total energy of the electron capture plus gamma would seem to be sufficient to couple with the positron decay. Also, given the note above (Line 70, Figure 1), is this argument affected by whether the positron decay goes directly to ground state or has an associated gamma?*

(lines 103 – 110) The energy that dictates if positron emission is possible is the Q value. The Q value is the difference between the initial mass state and final mass product. This energy is shared between the outgoing neutrino, atomic excitation of the daughter system, recoil energy, and possibly nuclear excitation of the daughter system. The Q-value therefore includes the excitation of the daughter system. However, positron decay can only compete with the electron capture if the Q value of the electron capture decay itself is greater than the threshold 1022 MeV value requiring the positron decay to go directly to the ground state. We will amend this to provide a clear statement in a revised manuscript.

3) *Line 106: For readers who are not nuclear physicists, a brief explanation of quantum selection rules would make this more readable.*

*(lines 121 - 123)* Quantum selection rules place formal constraints of the possible transitions of a system from one quantum state to another. In our case it places constraints on the possible set of transitions from the parent $^{40}$K state to the daughter $^{40}$Ar. We will include this definition in the revised manuscript.

4) *Figure 2 caption: Perhaps note that uncertainties were either not estimated or are smaller than the symbols? This is stated in the text but would be ideal to have in the caption as well.*

(Figure 2 caption) We will indicate in the caption that the uncertainties are either unknown or too small to plot.

5) *Line 287 (and throughout): The use of 'flux monitor' is a common error – should be 'fluence monitor', as they are measuring the total neutron fluence (flux over time) af- fecting samples over the entire irradiation, rather than monitoring the neutron flux at one specific time.*

(Lines 300, 320, 322, 333) We will correct flux monitor to fluence monitor.

6) *Line 303: Based on Figure 5, I calculate a different percent decrease for K-Ar ages at 1 Ga (2.5%, for ca. 25 Ma at 1 Ga, rather 1.3%). The value 0.7% at 4.5 Ga seems accurate, and the 1.6% at 1 Ma is not identifiable (due to scaling of graph) in the figure. It would be helpful to have an expanded Figure 5, with multiple scales (or just expand this scale down to 1 Ma) to highlight different parts of the geologic timescale. It would also be helpful to show results in relative values as well as absolute values. Finally, the K-Ar line seems to have structure (e.g. around 10^8 a) that should be explained.*

(line 319 – 340) For a $^{40}$Ar/$^{40}$K = 0.08, and using the decay constants in our Table 1 ($\lambda_{EC*}$ = 0.580e-10 and $\lambda_{total}$ = 5.463e-10 for Min et al; and $\lambda_{EC*}$ = 0.590e-10 and $\lambda_{total}$ = 5.473e-10; ), we used the K-Ar equation:

$$time = \frac{1}{\lambda_{total}} * \ln\left(1 + \frac{\lambda_{total}}{\lambda_{Ar}} \; or \; \frac{EC*^{40}Ar}{^{40}K}\right)$$

This yields dates of 1028.05 Ma and 1014.24 Ma for the Min et al. and our revised decay constants, respectively. This is a difference of 13.81 Ma, or about 1.3 %. We have reproduced this calculation and believe that our original calculation is correct, though we welcome any correction from AR2 if we have misunderstood.

In a revised manuscript, we will provide a figure that provides more detail. The small kink at 100 Ma is an Illustrator artifact and will be smoothed in a revised manuscript.

7) *Line 306: I'm struggling to understand the use of a fluence monitor at 23.2 Ma. I realize this is (may be?) a theoretical monitor, but it's in a paragraph with clear reference to Fish Canyon sanidine and I've spent some time wondering if '23.2' was a typo for '28.2 Ma'. Actually, I'm still not sure – is this a typo? If not, perhaps just note that it's a theoretical monitor of arbitrarily chosen age (if that's indeed what it is), to prevent others from wondering the same.*

(line 322- 323) we will change the date to 28.2 Ma.

Technical Corrections:

1) *Line 97: "They are linked because both processes have the same initial and final nu- clear states." It's not clear what 'they' refers to – likely electron capture and positron, but perhaps beta?*

(Line 99) "They" refers to EC and positron, and will be clarified in the revised text.

2) *Line 124: The symbol is missing from the pdf for type of emission*

(line 137) The missing quantity is a beta, and we will ensure this is typeset correctly in a revised text.

3) *Line 148: Is Emax defined somewhere?*

(line 164 - 165) Unfortunately we neglected to define $E_{max}$, but will do so in a revised text.

**Relevant changes to manuscript**

(1) The title has been modified to stress the implications of this decay mode for K-Ar dating

(2) Figure 2 has been changed to remove the preferred value.

(3) Figure 4 has been amended to include the Nähle et al. (2008) reference as suggested by reviewer 1.

(4) Table 1 has been changed to include modified decay constants using a lower and upper bound value for the electron capture to positron ratio of the ground state decay.

(5) Figure 5 has been changed to a 2 panel figure with the left panel showing both 
[revised manuscript text omitted]
 modelling, Lithos, 330, 120-138, http://dx.doi.org/10.1016/j.lithos.2019.02.007, 2019.

Alvarez, L.W., Nuclear K electron capture, Phys. Rev., 52(2), 134-135, http://dx.doi.org/10.1103/PhysRev.52.134 , 1937.

Audi, G., Bersillon, O., Blachot, J. and Wapstra, A.H., The NUBASE evaluation of nuclear and decay properties, Nucl. Phys. A, 729(1), 3-128, http://dx.doi.org/10.1016/j.nuclphysa.2003.11.001, 2003.

Baerg, A.P., Electron capture to positron branching ratios in the decay of $^{22}$Na and $^{44}$Sc, Can. J. Phys., 61(8), 1222-1226, http://dx.doi.org/10.1139/p83-155, 1983.

Bahcall, J.N., Electron Capture and Nuclear Matrix Elements of Be$^7$, Phys. Rev., 128(3), 1297-1301, http://dx.doi.org/10.1201/9780429502811-65, 1962.

Bambynek, W., Behrens, H., Chen, M.H., Crasemann, B., Fitzpatrick, M.L., Ledingham, K.W.D., Genz, H., Mutterer, M. and Intemann, R.L., Orbital electron capture by the nucleus, Rev. Mod. Phys., 49(1), 77-221, http://dx.doi.org/10.1103/RevModPhys.49.77, 1977.

Bé, M.-M., Chisté, V., Dulieu, C., Mougeot, X., Browne, E., Chechev, V., Kuzmenko, N., Kondev, F., Luca, A., Galan, M., Table of radionuclides (Vol. 5-A= 22 to 244), Mongraphie BIPM-5, https://hal-cea.archives-ouvertes.fr/cea-02476352/document, 2010.

Beckinsale, R.D. and Gale, N.H., A reappraisal of the decay constants and branching ratio of $^{40}$K, Earth Planet. Sci. Lett., 6(4), 289-294, http://dx.doi.org/10.1016/0012-821X(69)90170-8, 1969.

Bethe, H.A. and Bacher, R.F., Nuclear physics A. Stationary states of nuclei, Rev. Mod. Phys., 8(2), .82-229, http://dx.doi.org/10.1103/RevModPhys.8.82, 1936.

Bosch, H.E., Davidson, J., Davidson, M. and Szybisz, L., The electron capture to positron emission ratios in the decay of [22]Na and [65]Zn, Z. Phys. A., 280(4), 321-327, http://dx.doi.org/10.1007/BF01435440, 1977.

Chen, J., Nuclear Data Sheets for A=40. Nuclear Data Sheets 140, 1–376, http://dx.doi.org/10.1016/j.nds.2017.02.001, 2017.

Cresswell, A.J., Carter, J., Sanderson, D.C.W., Dose rate conversion parameters: Assessment of nuclear data, Radiat. Meas., 120, 195–201, http://dx.doi.org/10.1016/j.radmeas.2018.02.007, 2018.

Cresswell, A.J., Sanderson, D.C.W., Carter, J., Review of nuclear data for naturally occurring radionuclides applied to environmental applications, Eur. Phys. J. Plus, 134(2), 69, http://dx.doi.org/10.1140/epjp/i2019-12437-1, 2019.

Di Stefano, P.C.F., Brewer, N., Fijałkowska, A., Gai, Z., Goetz, K.C., Grzywacz, R., Hamm, D., Lechner, P., Liu, Y., Lukosi, E. and Mancuso, M., The KDK (potassium decay) experiment. arXiv [preprint], arXiv:1711.04004, 10 November 2017.

Di Stefano, P.C.F., Brewer, N., Fijałkowska, A., Gai, Z., Goetz, K.C., Grzywacz, R., Hamm, D., Lechner, P., Liu, Y., Lukosi, E. and Mancuso, M., The KDK (potassium decay) experiment. J. Phys. Conf. Ser. (Vol. 1342, No. 1, 012062). IOP Publishing, http://dx.doi.org/10.1088/1742-6596/1342/1/012062, 2020.

Emery, G.T., Ionization through Nuclear Electron Capture and Internal Conversion, in: Atomic Inner-Shell Processes, Academic Press, New York, 201–231, http://dx.doi.org/10.1016/B978-0-12-196901-1.50010-8, 1975.

Engelkemeir, D.W., Flynn, K.F. and Glendenin, L.E., Positron Emission in the Decay of K[40], Phys. Rev., 126(5), 1818-1822, http://dx.doi.org/10.1103/PhysRev.126.1818, 1962.

Endt, P.M., Energy levels of A = 21–44 nuclei (VII), Nucl. Phys. A 521, 1–400, http://dx.doi.org/10.1016/0375-9474(90)90598-G, 1990.

Endt, P.M. and Van der Leun, C., Energy levels of A= 21− 44 nuclei (V). Nucl. Phys. A, 214, 1-625, http://dx.doi.org/10.1016/0375-9474(73)91131-7, 1973.

Endt, P.M. and Van der Leun, C., Energy levels of A= 21–44 nuclei (VI). Nucl. Phys. A, 310(1-2), 1-751, http://dx.doi.org/10.1016/0375-9474(78)90611-5, 1978,

ENSDF Collaboration, LOGFT, http://www.nndc.bnl.gov/nndcscr/ensdf_pgm/ analysis/logft/unx/.

Fermi, E., Versuch einer Theorie der β-Strahlen. I, Z. Phys., 88, 161-177, http://dx.doi.org/10.1007/BF01351864, 1934.

Fireman, E.L., On the Decay of K[40], Phys. Rev., 75(9), 1447, http://dx.doi.org/10.1103/PhysRev.75.1447.2, 1949.

Forbes, G.B., Gallup, J. and Hursh, J.B., Estimation of total body fat from potassium-40 content, Science, 133(3446), 101-102., http://dx.doi.org/10.1126/science.133.3446.101, 1961.

Garner, E.L., Murphy, T.J., Gramlich, J.W., Paulsen, P.J., Barnes, I.L., Absolute isotopic abundance ratios and the atomic weight of a reference sample of potassium, J. Res. Natl. Bur. Stand. A Phys. Chem.: Vols., 79A(6), 713-725, http://dx.doi.org/10.6028/jres.079A.028, 1975.

Huber, P., Determination of antineutrino spectra from nuclear reactors, Phys. Rev. C, 84(2), 024617, http://dx.doi.org/10.1103/PhysRevC.84.024617, 2011.

Kossert, K. and Günther, E., LSC measurements of the half-life of [40]K, Appl. Radiat. Isot., 60(2-4), 459-464, http://dx.doi.org/10.1016/j.apradiso.2003.11.059, 2004.

Krane, K.S., Halliday, D., 3rd ed., Introductory nuclear physics, Wiley, New York, 845pp., 1987.

Kreger, W.E., K Capture to positron ratio for Na[22], Phys. Rev., 96(6), 1554-1555, http://dx.doi.org/10.1103/PhysRev.96.1554, 1954.

Kuiper, K.F., Deino, A., Hilgen, F.J., Krijgsman, W., Renne, P.R., Wijbrans, J.R., Synchronizing rock clocks of Earth history, Science, 320(5875), 500-504, http://dx.doi.org/10.1126/science.1154339, 2008.

Kunze, V., Schmidt-Ott, W.D. and Behrens, H., Remeasurement of capture to positron decay ratios in [22]Na and [65]Zn and comparison with theory, Z. Phys. A, 337(2), 169-173, http://dx.doi.org/10.1007/BF01294288, 1990.

Leutz, H., Schulz, G. and Wenninger, H., The decay of potassium-40, Z. Phys., 187(2), 151-164, http://dx.doi.org/10.1007/BF01387190, 1965.

MacMahon, T.D. and Baerg, A.P., The electron capture to positron branching ratio in the decay of [22]Na, Can. J. Phys., 54(14), 1433-1437, http://dx.doi.org/10.1139/p76-168, 1976.

Malonda, A.G. and Carles, A.G., Half-life determination of [40]K by LSC, Appl. Radiat. Isot., 56(1-2), 153-156, http://dx.doi.org/10.1016/S0969-8043(01)00181-6, 2002.

Marshall, B.D., DePaolo, D.J., Precise age determinations and petrogenetic studies using the K-Ca method, Geochim. Cosmochim. Acta, 46(12), 2537–2545, http://dx.doi.org/10.1016/0016-7037(82)90376-3, 1982.

McCann, M.F., Smith, K.M., Direct measurement of the K electron capture to positron emission ratio in the decay of [22]Na, J. Phys. A Gen. Phys., 2(3), 392–397, http://dx.doi.org/10.1088/0305-4470/2/3/018, 1969.

McDougall, I., Harrison, T.M., 2nd ed., Geochronology and Thermochronology by the [40]Ar/[39]Ar Method, Oxford University Press, Oxford. 269, 1999.

Merrihue, C. and Turner, G., Potassium-argon dating by activation with fast neutrons, J. Geophys. Res. Solid Earth, 71(11), 2852-2857, http://dx.doi.org/10.1029/JZ071i011p02852, 1966.

Min, K., Mundil, R., Renne, P.R. and Ludwig, K.R., A test for systematic errors in [40]Ar/[39]Ar geochronology through comparison with U/Pb analysis of a 1.1-Ga rhyolite, Geochim. Cosmochim. Acta, 64(1), 73-98, http://dx.doi.org/10.1016/S0016-7037(99)00204-5, 2000.

Morgan, L.E., Mark, D.F., Imlach, J., Barfod, D. and Dymock, R., FCs-EK: A new sampling of the Fish Canyon Tuff 40Ar/39Ar neutron flux monitor, Geol. Soc. Spec. Publ., 378(1), 63-67, http://dx.doi.org/10.1144/SP378.21, 2014.

Mougeot, X., Helmer, R.G., [40]K – Comments on evaluation of decay data, LNHB/ INEEL, http://www.nucleide.org/DDEP_WG/Nuclides/K-40_com.pdf, 2009.

Mougeot, X., Improved calculations of electron capture transitions for decay data and radionuclide metrology, Appl. Radiat. Isot., 134, 225-232, http://dx.doi.org/10.1016/j.apradiso.2017.07.027, 2018.

Nähle, O., Kossert, K. and Klein, R., Activity standardization of 22Na, Appl. Radiat. Isot., 66(6-7), 865-871, http://dx.doi.org/10.1016/j.apradiso.2008.02.028, 2008.

Patterson, C., Age of meteorites and the earth, Geochim. Cosmochim. Acta, 10(4), 30-237, http://dx.doi.org/10.1016/0016-7037(56)90036-9, 1956.

Pradler, J., Singh, B. and Yavin, I., On an unverified nuclear decay and its role in the DAMA experiment, Phys. Lett. B, 720(4-5), 399-404, http://dx.doi.org/10.1016/j.physletb.2013.02.033, 2013.

Preece, K., Mark, D.F., Barclay, J., Cohen, B.E., Chamberlain, K.J., Jowitt, C., Vye-Brown, C., Brown, R.J. and Hamilton, S., Bridging the gap: $^{40}Ar/^{39}Ar$ dating of volcanic eruptions from the 'Age of Discovery', Geology, 46(12), 1035-1038, http://dx.doi.org/10.1130/G45415.1, 2018.

Renne, P.R., Sharp, W.D., Deino, A.L., Orsi, G. and Civetta, L., $^{40}Ar/^{39}Ar$ dating into the historical realm: Calibration against Pliny the Younger, Science, 277(5330), 1279-1280, http://dx.doi.org/10.1126/science.277.5330.1279, 1997.

Renne, P.R., Swisher, C.C., Deino, A.L., Karner, D.B., Owens, T.L. and DePaolo, D.J., Intercalibration of standards, absolute ages and uncertainties in $^{40}Ar/^{39}Ar$ dating, Chem. Geol., 145(1-2), 117-152, http://dx.doi.org/10.1016/S0009-2541(97)00159-9, 1998.

Renne, P.R., $^{40}Ar/^{39}Ar$ age of plagioclase from Acapulco meteorite and the problem of systematic errors in cosmochronology. Earth Planet. Sci. Lett., 175(1-2), 13-26, http://dx.doi.org/10.1016/S0012-821X(99)00287-3, 2000.

Renne, P.R., Mundil, R., Balco, G., Min, K. and Ludwig, K.R., Joint determination of $^{40}K$ decay constants and $^{40}Ar*/^{40}K$ for the Fish Canyon sanidine standard, and improved accuracy for $^{40}Ar/^{39}Ar$ geochronology, Geochim. Cosmochim. Acta, 74(18), 5349-5367, http://dx.doi.org/10.1016/j.gca.2010.06.017, 2010.

Renne, P.R., Balco, G., Ludwig, K.R., Mundil, R., Min, K., Response to the comment by W.H. Schwarz et al. on "Joint determination of $^{40}K$ decay constants and $^{40}Ar*/^{40}K$ for the Fish Canyon sanidine standard, and improved accuracy for $^{40}Ar/^{39}Ar$ geochronology" by P.R. Renne et al. (2010), Geochim. Cosmochim. Acta, 75, 5097–5100, http://dx.doi.org/10.1016/j.gca.2011.06.021, 2011.

Rivera, T.A., Storey, M., Zeeden, C., Hilgen, F.J., Kuiper, K., A refined astronomically calibrated $^{40}Ar/^{39}Ar$ age for Fish Canyon sanidine, Earth Planet. Sci. Lett., 311, 420–426, http://dx.doi.org/10.1016/j.epsl.2011.09.017, 2011.

Schmidt-Ott, W.-D., Lauerwald, J., Bosch, U., Dornh6fer, H., Schrewe, U.J., Behrens, H., Electron-capture to positron ratio in the decays of $^{22}Na$ and $^{65}Zn$: Proceedings of the 7th International Conference on Atomic Masses and Fundamental Constants AMCO-7, Darmstadt: Technische Hochschule Darmstadt, Lehrdruckerei, 3- 7 September 1984.

Steiger, R. and Jäger, E., Subcommission on geochronology: convention on the use of decay constants in geo-and cosmochronology, Earth Planet. Sci. Lett., 36(3), 359-362, http://dx.doi.org/10.1016/0012-821X(77)90060-7, 1977.

Stukel, M.: Characterization Of Large Area Avalanche Photodiodes For The Measurement Of The Electron Capture Decay Of $^{40}$K To The Ground State Of $^{40}$Ar. M.Sc. Thesis, Queen's University. 159 pp., 2018.

Sýkora, I. and Povinec, P., Measurement of electron capture to positron emission ratios in light and medium nuclides, Nucl. Instrum. Methods Phys. Res. B, 17(5-6), 467-471, http://dx.doi.org/10.1016/0168-583X(86)90189-8, 1986.

Vatai, E., Varga, D. and Uchrin, J., Measurement of the $\epsilon/\beta+$ ratio in the decay of $^{22}$Na and $^{74}$As, Nucl. Phys. A, 116(3), 637-642, http://dx.doi.org/10.1016/0375-9474(68)90396-5, 1968.

Wang, M., Audi, G., Kondev, F.G., Huang, W.J., Naimi, S., Xu, X., The AME2016 atomic mass evaluation (II). Tables, graphs and references, Chinese Phys. C 41(3), 030003, https://doi.org/10.1088/1674-1137/41/3/030003, 2017.

Wasserburg, G.J. and Hayden, R.J., A$^{40}$-K$^{40}$ dating. Geochim. Cosmochim. Acta, 7(1-2), 51-60, http://dx.doi.org/10.1016/0016-7037(55)90045-4, 1955.

Williams, A., Measurement of the ratio of electron capture to positon emission in the decay of Na-22, Nucl. Phys., 52, 324-332, http://dx.doi.org/10.1016/0029-5582(64)90696-0, 1964.

Yukawa, H. and Sakata, S., On the Theory of the β-Disintegration and the Allied Phenomenon, Proc. Phys. Math. Soc. Jpn., 3rd Series, 17, 467-479, https://www.jstage.jst.go.jp/article/ppmsj1919/17/0/17_0_467/_pdf, 1935.